# Identification of Hub Genes and Key Pathways Associated with Sepsis Progression Using Weighted Gene Co-Expression Network Analysis and Machine Learning

**DOI:** 10.3390/ijms26094433

**Published:** 2025-05-07

**Authors:** Qinghui Sun, Hai-Li Zhang, Yichao Wang, Hao Xiu, Yufei Lu, Na He, Li Yin

**Affiliations:** 1School of Tropical Medicine, Hainan Medical University, Haikou 571199, China; hy0211068@hainmc.edu.cn (Q.S.); hy0211064@muhn.edu.cn (H.X.); luackroyd@outlook.com (Y.L.); hy0211078@muhn.edu.cn (N.H.); 2NHC Key Laboratory of Tropical Disease Control, School of Tropical Medicine, Hainan Medical University, Haikou 571199, China; 3School of Environment and Geography, Qingdao University, Qingdao 266071, China; zhanghaili@qdu.edu.cn (H.-L.Z.); wangyichaoqdu@163.com (Y.W.)

**Keywords:** sepsis, transcriptomic analysis, weighted gene co-expression network analysis, biomarkers, pathway enrichment, TNFSF10, TMCC2, PLVAP

## Abstract

Sepsis is a life-threatening condition driven by dysregulated immune responses, resulting in organ dysfunction and high mortality rates. Identifying key genes and pathways involved in sepsis progression is crucial for improving diagnostic and therapeutic strategies. This study analyzed transcriptomic data from 49 samples (37 septic patients across days 0, 1, and 8, and 12 healthy controls) using weighted gene co-expression network analysis (WGCNA) and multi-algorithm feature selection approaches. Differential expression analysis, pathway enrichment, and network analyses were employed to uncover potential biomarkers and molecular mechanisms. WGCNA identified modules such as MEbrown4 and MEblack, which strongly correlated with sepsis progression (r > 0.7, *p* < 0.01). Differential expression analysis highlighted up-regulated genes like CD177 and down-regulated genes like LOC440311. KEGG analysis revealed significant pathways including neuroactive ligand–receptor interaction, PI3K-Akt signaling, and MAPK signaling. Gene ontology analysis showed involvement in immune-related processes such as complement activation and antigen binding. Protein–protein interaction network analysis identified hub genes such as TNFSF10, IGLL5, BCL2L1, and SNCA. Feature selection methods (random forest, LASSO regression, SVM-RFE) consistently identified top predictors like TMCC2, TNFSF10, and PLVAP. Receiver operating characteristic (ROC) analysis demonstrated high predictive accuracy for sepsis progression, with AUC values of 0.973 (TMCC2), 0.969 (TNFSF10), and 0.897 (PLVAP). Correlation analysis linked key genes such as TNFSF10, GUCD1, and PLVAP to pathways involved in immune response, cell death, and inflammation. This integrative transcriptomic analysis identifies critical gene modules, pathways, and biomarkers associated with sepsis progression. Key genes like TNFSF10, TMCC2, and PLVAP genes show strong diagnostic potential, providing novel insights into sepsis pathogenesis and offering promising targets for future therapeutic interventions. Among these, TNFSF10 and PLVAP are known to encode secreted proteins, suggesting their potential as circulating biomarkers. This enhances their translational relevance in clinical diagnostics.

## 1. Introduction

Sepsis, a concerning infectious disorder, remains a significant issue within modern medicine. It is an inflammatory response instigated by microbes like bacteria, fungi, or viruses invading the blood system, typically leading to elevated death rates [1]. Often, it leads to malfunctioning organs, shock, and multi-organ failure, significantly impacting patient well-being [2]. In particular, among senior citizens, due to factors like waning immunity and multiple chronic ailments, the risk of acquiring sepsis and succumbing to it is amplified. Hence, exploring and managing elderly sepsis holds immense clinical importance [3].

With the development of bioinformatics and systems biology, methods such as weighted gene co-expression network analysis (WGCNA) are among the important tools for studying the mechanisms of complex diseases [4,5].

This study aims to utilize WGCNA to explore the hub genes and key pathways that play pivotal roles in sepsis progression. By analyzing gene expression data from sepsis patients, we seek to shed light on the molecular signatures that underlie the worsening of this critical condition. The findings of this research may provide valuable insights for the development of novel therapeutic interventions and personalized treatment approaches for sepsis patients.

## 2. Result

### 2.1. Data Collection and Preprocessing

Data for a sequence expression matrix were obtained from NCBI for 49 samples. This file portrayed 37 septic patients (day 0, day 1, and day 8) along with 12 healthy controls. Accordingly, the requisite input files for WGCNA were created: an expression file and a feature file.

### 2.2. Construction of Weighted Gene Co-Expression Network of the Sepsis and Normal

This study employs WGCNA to explore gene modules associated with the severity of sepsis. A heatmap of significantly differentially expressed genes is presented, revealing distinct hierarchical clustering patterns between control (C) and sepsis (P) samples. Genes with elevated expression in sepsis are shown in red, while down-regulated genes are indicated in blue (Figure 1A). The volcano plot illustrates the distribution of genes based on log2 fold-change (logFC) and statistical significance (−log10 *p*-value), highlighting key differentially expressed genes such as *CD177* (up-regulated) and *LOC440311* (down-regulated) in red and blue, respectively (Figure 1B). The scale-free topology model fit (R^2^) under varying soft-threshold powers identifies a threshold of approximately 8 as optimal for achieving scale-free properties. The accompanying mean connectivity plot confirms that this threshold preserves network integrity while avoiding overfitting (Figure 1C). A sample dendrogram groups samples by transcriptomic similarity, while a trait heatmap below correlates these groups (control, day 0, day 1, day 8) with clinical traits (Figure 1D). The clustering of genes into co-expression modules is depicted, with dynamic tree cutting assigning distinct module colors to capture co-expression patterns (Figure 1E). Notably, modules such as MEbrown4 and MEblack exhibit strong positive correlations with sepsis progression (r > 0.7, *p* < 0.01), suggesting them as potential targets for further exploration into the molecular mechanisms of sepsis (Figure 1F).

The Venn diagram shows 54 overlapping genes identified through WGCNA and differential expression analysis, highlighting their potential relevance in sepsis progression (Figure 2A). KEGG pathway enrichment analysis revealed significant pathways, such as “Neuroactive ligand-receptor interaction”, “PI3K-Akt signaling”, and “MAPK signaling”, highlighting their roles in sepsis-related biological processes (Figure 2B). Gene ontology (GO) analysis showed that enriched genes are involved in immune-related processes, including “complement activation (BP)”, “immunoglobulin complex (CC)”, and “antigen binding (MF)”, highlighting their roles in sepsis progression (Figure 2C). A PPI network identified key hub genes, including *IGLL5*, *BCL2L1*, *TNFSF10*, and *SNCA*, with significant connections, highlighting their critical roles in sepsis development and potential as biomarkers or therapeutic targets (Figure 2D).

### 2.3. Multi-Algorithm Feature Selection for Identifying Key Genes in Sepsis Progression

The Venn diagram compares the results of the three feature selection methods: random forest, LASSO, and SVM-RFE (Figure 3A). The left panel of Figure 3B highlights the top-ranked genes based on feature importance from the random forest algorithm. Genes such as *TMC2*, *TNFSF10*, and *CTNNA1* were the most significant predictors of sepsis severity. The right panel shows the random forest error rate across iterations, demonstrating convergence and stability with increasing trees. The left plot of Figure 3C displays the trajectories of gene coefficients as the regularization parameter **log(λ)** changes. Important genes are selected as their coefficients deviate from zero. The right plot shows the binomial deviance across different **λ** values, with the optimal **λ** minimizing deviance, indicating the best-performing gene set selected by LASSO. The left panel of Figure 3D depicts cross-validated accuracy as a function of the number of selected features, demonstrating that accuracy peaks when 10 features are included. The right panel plots cross-validated error, which decreases significantly with feature selection, further supporting the robustness of these genes in predicting sepsis progression.

### 2.4. Key Genes Identified in Sepsis Progression

A prediction nomogram was constructed to assess the contributions of key genes to sepsis progression. Genes including *TMCC2*, *TNFSF10*, *TFDP1*, *BNIP3L*, *PLVAP*, *ESPN*, *MRC2*, *GUCD1*, *UBA52*, and *EPB41* are displayed with their respective scores contributing to the overall total points (Figure 4A). Receiver operating characteristics show that key genes such as *TMCC2* (AUC = 0.973), *TNFSF10* (AUC = 0.969), and *PLVAP* (AUC = 0.897) demonstrate high predictive accuracy for sepsis progression, underscoring their potential as diagnostic biomarkers (Figure 4B).

### 2.5. Identification of Key Genes in Sepsis Progression Using Transcriptomic Analysis

The bar chart compares pathway enrichment scores between control (C, green) and sepsis patients (P, red), revealing significant differences in pathways related to immune response, cell death, and inflammation (marked with asterisks), suggesting their critical roles in sepsis progression (Appendix A). The heatmap (Figure 5A) illustrates the correlations between enriched pathways and key genes, with red indicating positive correlations and green representing negative correlations. Notably, *TNFSF10*, *GUCD1*, and *PLVAP* show strong associations with pathways sepsis-relevant sepsis development. The correlation matrix (Figure 5B) highlights strong positive relationships among key genes such as *TNFSF10*, *EPB41*, *PLVAP*, and *TMCC2*, suggesting their coordinated involvement in sepsis pathogenesis. Boxplots (Figure 5C) compare the expression levels of key genes (*EPB41*, *UBA52*, *GUCD1*, *MRC2*, *PLVAP*, *BNIP3L, TFDP1*, *TNFSF10*, and *TMCC2*) between control and sepsis patients, with most genes showing significantly higher expression in sepsis patients, highlighting their potential as biomarkers or therapeutic targets.

## 3. Material and Methods

### 3.1. Data Processing

RNA-Seq datasets GSE216902 and GSE232753 were obtained from the Gene Expression Omnibus (GEO) database [1,6]. GSE216902 consists of 37 whole blood samples (31 sepsis, 6 healthy controls), while GSE232753 includes 12 PBMC samples (6 sepsis, 6 controls). For each dataset, raw count matrices were processed using the DESeq2 package in R [7]. Genes with fewer than 10 counts were filtered out, and expression values were normalized using size factors, followed by log2 transformation. Genes with a maximum expression < 5 were excluded. To integrate the two datasets, we first extracted the intersection of shared genes, standardized expression within each dataset, and then applied batch effect correction using the ComBat algorithm from the sva R package. This approach accounted for differences in sample types (whole blood vs. peripheral blood mononuclear cells (PBMCs)) while preserving biologically meaningful variation. The resulting expression matrix was used for downstream WGCNA.

### 3.2. Weighted Gene Coexpression Network Analysis (WGCNA)

The application of WGCNA aimed to uncover the intricate regulatory associations inherent in the gene expression datasets [8]. Leveraging the R programming language, complemented by pertinent libraries and packages, streamlined the computational analyses. Initial quality assessments were conducted to evaluate the integrity of both samples and genes. Identification of outliers ensued through hierarchical clustering of samples. The seamless integration of clinical trait information with gene expression data allowed for the exploration of correlations between clinical traits and sample clustering [9]. To attain a scale-free topology, soft-thresholding powers were meticulously selected. The resulting adjacency matrix underwent transformation into a topological overlap matrix (TOM). The construction of gene dendrograms and the identification of modules employed dynamic tree cutting. Evaluation of modules with akin expression profiles led to the computation of eigengenes. Subsequent clustering based on eigengene correlation facilitated the merger of modules demonstrating notable similarity, paving the way for comprehensive downstream analysis.

In the exploration of gene co-expression networks, we systematically identified pivotal genes within each module by focusing on those with high values simultaneously for both “kTotal” and “kWithin”, which are network connectivity measures calculated by the WGCNA R package. Specifically, “kTotal” represents the total connectivity of a gene across the entire network, while “kWithin” reflects its intramodular connectivity. To select highly connected genes, we established thresholds based on percentiles, such as selecting genes within the upper 70% range. For each module, the respective percentiles for both parameters were computed, and genes exceeding both thresholds were designated as candidate hub genes.

### 3.3. Gene Ontology (GO) Enrichment Analysis

The Gene Ontology (GO) database, developed by the Gene Ontology Consortium, provides a standardized framework for describing gene and protein functions across species. It consists of three main categories: Biological Process (BP), Cellular Component (CC), and Molecular Function (MF). GO enrichment results were obtained using Bioinformatics (https://www.bioinformatics.com.cn, accessed on 29 November 2024).

### 3.4. Kyoto Encyclopedia of Genes and Genomes (KEGG)

The Kyoto Encyclopedia of Genes and Genomes (KEGG) database provides a comprehensive resource for understanding high-level functions and utilities of biological systems, such as metabolism, cellular processes, and organismal systems, based on molecular-level information. It systematically integrates gene functions, genomic information, and biochemical pathways. In this study, enrichment analysis was performed by comparing the differentially expressed genes with curated KEGG pathway gene sets. The top 10 pathways with the lowest *p*-values were extracted and visualized. Pathways with *p* < 0.05 and log |fold change| > 0 were considered statistically significant. KEGG enrichment results were obtained using Bioinformatics (https://www.bioinformatics.com.cn, accessed on 29 November 2024).

### 3.5. Method for Disease Feature Gene Selection Using Three Machine Learning Classification Algorithms

In this study, three machine learning classification algorithms, support vector machine recursive feature elimination (SVM-RFE) [10], random forest (RF) [11], and least absolute shrinkage and selection operator (LASSO) [12], were employed to identify disease-related feature genes. Complementary machine learning methods, ensures a robust and reliable selection of disease-related feature genes.

### 3.6. Receiver Operating Characteristic (ROC) Curve Analysis

To evaluate the predictive performance of the selected genes, an ROC curve analysis was conducted [13]. The pROC package in R was used to generate ROC curves for the following genes: *EPB41*, *UBA52*, *GUCD1*, *MRC2*, *EVAP*, *BNIP3L*, *TFDP1*, *TNFSF10*, and *TMCC2*. The roc () function was applied with the clinical outcome variable (Type) as the response and the selected genes as predictors. The resulting ROC curves were visualized using the ggroc () function from the ggplot2 package, with color enhancements provided by the ggsci::scale_color_lancet () function. The area under the curve (AUC) values were assessed to determine the diagnostic accuracy of the biomarkers. A logistic regression model was built using the selected genes to predict the clinical outcome. The rms package in R was used for model fitting. The dataset aSAH was utilized, with the dependent variable being the clinical type (Type), and the independent variables being the selected genes (*EPB41*, *UBA52*, *GUCD1*, *MRC2*, *EVAP*, *BNIP3L*, *TFDP1*, *TNFSF10*, and *TMCC2*). The logistic regression model was fitted using the lrm () function.

### 3.7. Gene Set Enrichment Analysis (GSEA)

The getGmt () function from the GSEABase package was utilized to load the gene sets, which were associated with gene identifiers corresponding to gene symbols. To increase the accuracy of GSVA results, we employed a parameterization strategy with the gsvaParam () function, including the maxDiff parameter, which emphasizes the maximal differences in gene set expression between samples [14]. Pathways were considered significant if they showed a |normalized enrichment score (NES)| > 1.5 with an associated *p*-value < 0.05.

### 3.8. Statistical Analysis and Differential Expression

To assess the differential expression of selected genes between the control and disease groups, *t*-tests were applied. The ggpubr package in R was used to generate box plots for visualizing gene expression differences between the disease and control groups [15,16]. The ggboxplot () function was employed to create box plots for each selected gene, with comparisons between the two groups performed using the stat_compare_means () function. To ensure robust statistical comparisons, pairwise comparisons were conducted across all group combinations using the combn () function to generate the comparison groups. The ggsci package was used to customize the color palette, with the control group represented in blue (Ccol) and the disease group in red (Pcol).

## 4. Discussion

Sepsis is a critical syndrome characterized by dysfunctional organ responses triggered by a dysregulated host response [2]. It represents a significant healthcare challenge, impacting millions worldwide annually and resulting in the mortality of approximately 17–33% of affected individuals [17,18,19]. The incidence of sepsis and its associated healthcare burden are both escalating, attributed to global population aging trends and the rising prevalence of comorbidities among patients [19]. Developing biomarkers could play a pivotal role in sepsis management, potentially alleviating its healthcare burden. Early detection is paramount for prompt intervention and enhancing sepsis outcomes [20].

In this study, we combined WGCNA, differential expression analysis, and machine learning approaches to identify candidate biomarkers and elucidate molecular mechanisms of sepsis progression. Clustering of differentially expressed genes (DEGs) between control and sepsis samples showed significant differences, with significant up-regulation of *CD177* and down-regulation of *LOC440311*, which may reflect a shift from hyperinflammation to immunosuppression. *CD177* is a well-established marker of neutrophil activation and is closely associated with the hyperinflammatory phase of sepsis, consistent with previous studies indicating that neutrophil dysfunction contributes to systemic inflammation and organ failure [21]. Conversely, the down-regulation of *LOC440311* may indicate engagement of immunosuppression and metabolic pathways during the later stages of the disease.

Pathway analyses revealed critical involvement of immune and inflammatory signaling, with regulate apoptosis and cytokine responses in sepsis. For instance, the PI3K-Akt pathway regulates cellular survival and proliferation, which is dysregulated in septic tissues, contributing to immune cell apoptosis and organ dysfunction [22]. Similarly, MAPK signaling plays a role in cytokine production and inflammatory cascades, driving the uncontrolled immune response characteristic of sepsis [23]. GO enrichment highlighted excessive activation of complement and immune effector processes, potentially exacerbating tissue injury [24]. We identified hub genes within the PPI network, including-TNFSF10, *BCL2L1*, and *PLVAP*, which are closely linked to immune regulation and apoptotic processes. Notably, *TNFSF10* (*TRAIL*) plays a pivotal role in immune cell apoptosis and was consistently elevated in septic individuals [25]. Both *TNFSF10* and *PLVAP* are secreted or membrane-bound proteins detectable in circulation, underscoring their potential as clinically accessible biomarkers. *BCL2L1* encodes Bcl-xL, an anti-apoptotic protein that promotes cellular survival during stress responses, though its dysregulation may impair immune homeostasis in sepsis.

To refine candidate selection, we implemented machine learning algorithms (random forest and SVM-RFE) [10], which robustly identified *TMCC2*, *TNFSF10*, and *CTNNA1* as key predictors of sepsis severity. *TMCC2* and *TNFSF10* demonstrated strong diagnostic performance with AUCs of 0.973 and 0.969, respectively. Elevated *TNFSF10* expression was associated with increased immune cell apoptosis, a hallmark of the immunosuppressive phase of sepsis. PLVAP, with an AUC of 0.897, also showed strong predictive potential, supporting its role in vascular permeability and endothelial barrier dysfunction 

Further enrichment analysis of pathway activity scores confirmed differential regulation of immune response, cell death (apoptosis/necroptosis), and inflammation between septic and control samples. These biological processes are fundamental to sepsis progression, where uncontrolled inflammatory signaling drives the release of cytokines (e.g., interleukins and TNF-α) and programmed cell death contributes to immune cell depletion and organ failure. Excessive activation of neutrophils and macrophages during early phases promotes systemic inflammation, while the subsequent immunosuppressive state compromises pathogen clearance and increases susceptibility to secondary infections. Correlation analyses further demonstrated coordinated expression patterns among *TNFSF10*, *EPB41*, *PLVAP*, reflecting shared functions in immune regulation, endothelial integrity, and inflammatory signaling. For example, *EPB41* (erythrocyte membrane protein band 4.1), a cytoskeletal component, may influence leukocyte adhesion and transmigration, while TMCC2 has been implicated in cellular stress responses, including apoptosis and inflammation [26,27].

Expression profiling also confirmed the significant up-regulation of *EPB41*, *UBA52*, *GUCD1*, *MRC2*, *BNIP3L*, *and TFDP1* in septic patients. These genes are functionally associated with cytoskeletal remodeling, mitochondrial integrity, phagocytosis, and immune regulation [28], supporting their relevance as biomarkers and contributors to sepsis pathology.

Although the present study is limited by the lack of experimental validation, future work will incorporate qPCR, Western blotting, and functional assays to confirm the expression and biological roles of *TNFSF10*, *TMCC2*, and *PLVAP* in sepsis. In addition, we plan to perform cross-validation using independent transcriptomic datasets to further assess the robustness and generalizability of these gene signatures.

## 5. Conclusions

In conclusion, our integrative approach identified several robust gene signatures, including *TNFSF10*, *PLVAP*, and *TMCC2*, as potential biomarkers of sepsis severity. Notably, the detectability of TNFSF10 and PLVAP in plasma strengthens their translational relevance. These findings contribute to a more comprehensive understanding of sepsis biology and provide a foundation for developing targeted diagnostic and therapeutic strategies.

## Figures and Tables

**Figure 1 ijms-26-04433-f001:**
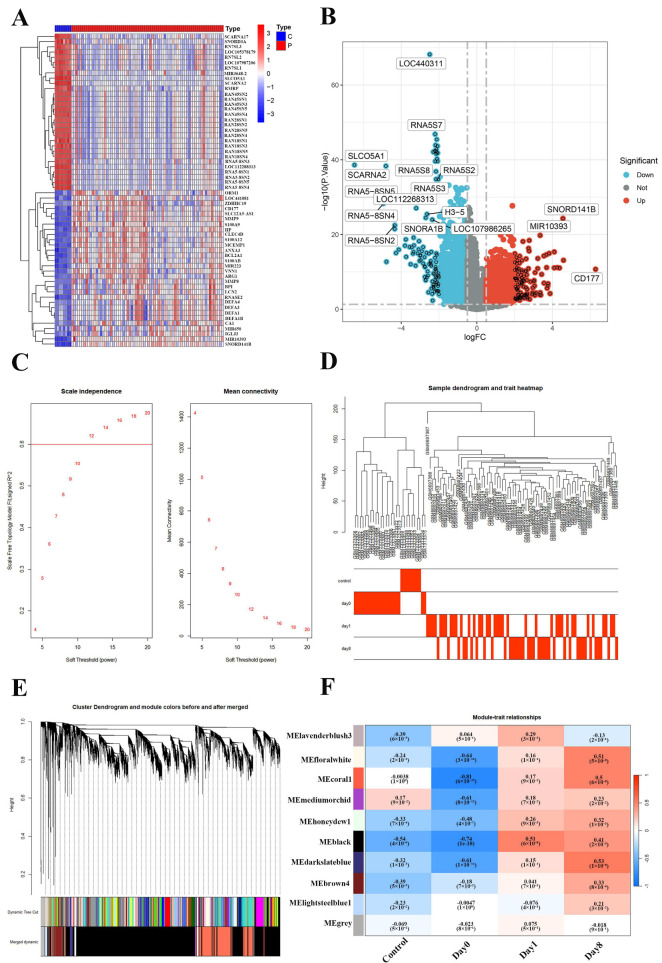
Determination of soft-threshold power in the WGCNA: (**A**) heatmap of differentially expressed genes (DEGs); (**B**) volcano plot of DEGs; (**C**) scale-free topology and soft threshold selection; (**D**) sample clustering and trait heatmap; (**E**) cluster dendrogram of genes based on topological overlap, with module colors assigned before (Dynamic Tree Cut) and after module merging (Merged dynamic), each color represents a distinct gene co-expression module; (**F**) moduletrait relationships.

**Figure 2 ijms-26-04433-f002:**
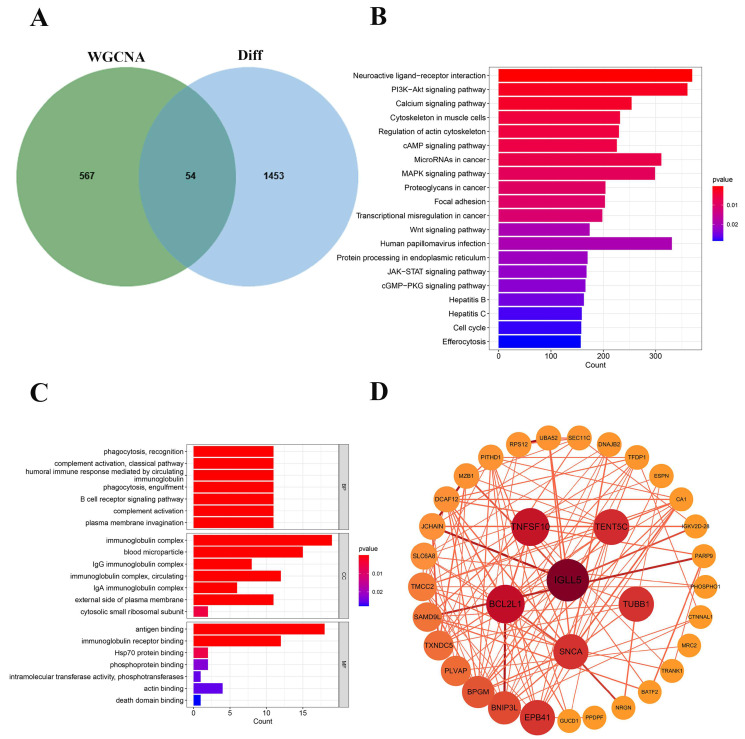
Functional pathway enrichment analysis: (**A**) venn diagram of gene overlap; (**B**) KEGG pathway enrichment analysis; (**C**) GO functional enrichment analysis; (**D**) protein-protein interaction (PPI) network analysis, node size and color intensity correspond to the degree of connectivity, with darker and larger nodes indicating higher connectivity.

**Figure 3 ijms-26-04433-f003:**
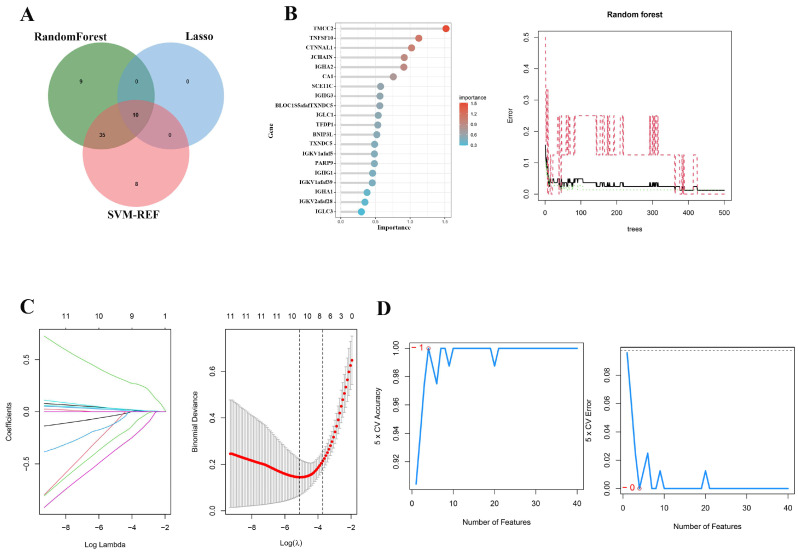
Identify critical genes involved in sepsis progression through feature selection methods: (**A**) Venn diagram of selected genes; (**B**) random forest feature importance and error curve. Top important genes ranked by the Random Forest model, darker red indicates higher importance (Left). Error rate curve of the Random Forest model across the number of trees, the black solid line represents the overall model error, while the red dashed lines represent the class-specific error rates (Right); (**C**) LASSO regression results. LASSO coefficient profiles of the candidate genes plotted against the log(λ) sequence, each colored line represents the trajectory of a single gene’s coefficient (Left), ten-fold cross-validation for tuning parameter selection in LASSO. The binomial deviance is plotted against log(λ), and the optimal λ values are indicated by the dotted vertical lines (Right); (**D**) SVM-RFE cross-validation results. Five-fold cross-validation accuracy plotted against the number of features selected (Left), cross-validation error plotted against the number of features selected (Right).

**Figure 4 ijms-26-04433-f004:**
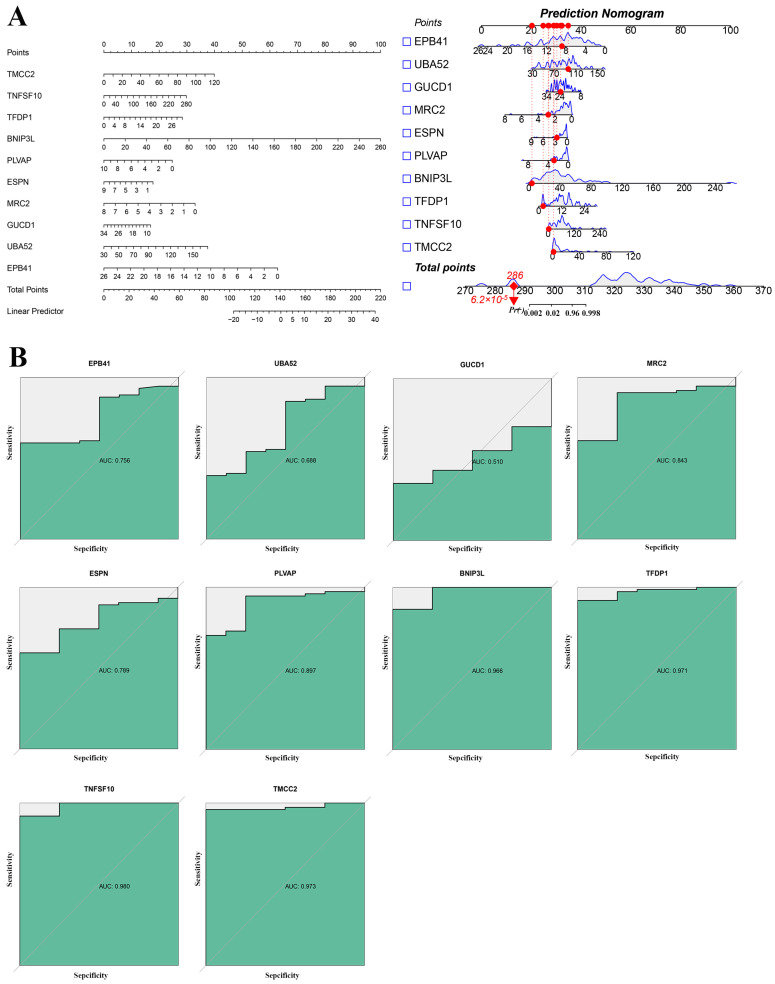
Predictive performance and nomogram for key genes in sepsis: (**A**) a nomogram illustrating the contribution of key genes to sepsis risk prediction. The point assignment for each variable (hub gene) in the nomogram model (Left). The calibration of the nomogram prediction, red dots represent the actual observed values at each point for individual variables, blue lines represent the estimated prediction distribution or probability curves for each gene and the total points (Right); (**B**) receiver operating characteristic curves.

**Figure 5 ijms-26-04433-f005:**
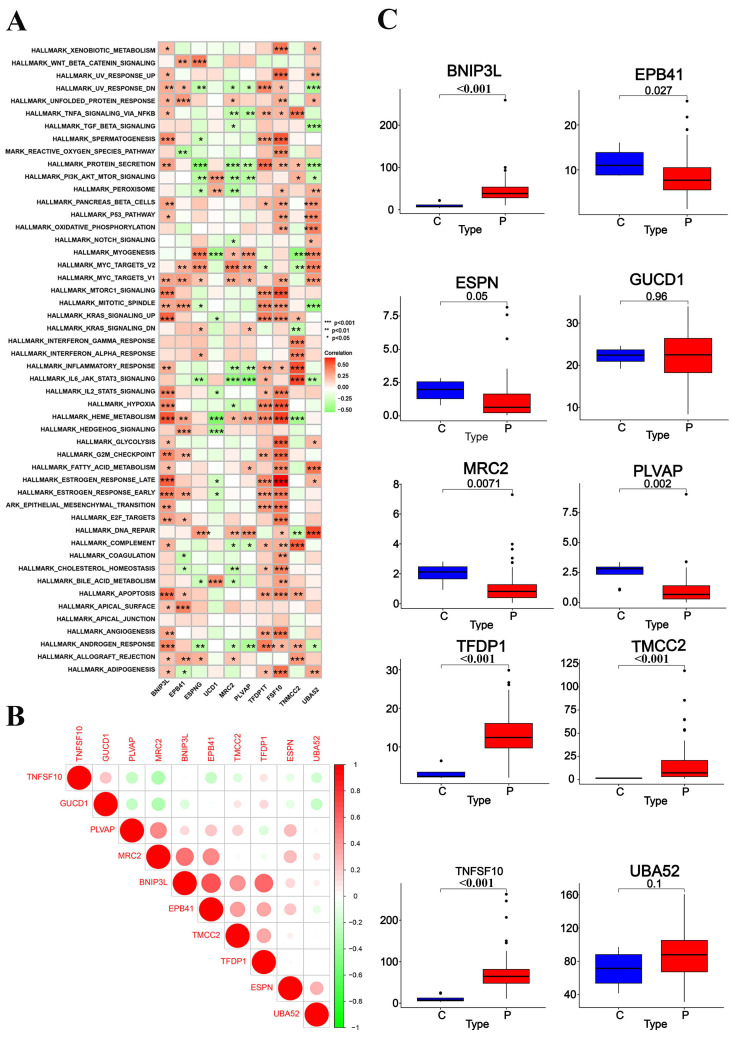
Pathway enrichment, correlation analysis, and gene expression differences between control and sepsis patients: (**A**) genomic enrichment analysis (GSEA), heatmap; (**B**) correlation matrix highlighting the relationships among key genes (circle size and color intensity represent the strength and direction of correlations, respectively); (**C**) boxplots comparing the expression levels of key genes between control (C, blue) and sepsis patients (P, red). Statistically significant differences are indicated by *p*-values.

## Data Availability

The RNA-Seq datasets analyzed in this study are publicly available in the GEO database under the accession numbers GSE216902 and GSE232753.

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
