# Peer review of "Identification of Hub Genes and Key Pathways Associated with Sepsis Progression Using Weighted Gene Co-Expression Network Analysis and Machine Learning"

_ijms, 2025, doi:10.3390/ijms26094433_

Round 1
Reviewer 1 Report
Comments and Suggestions for Authors
This work from Sun wet al utilizes Weighted Gene Co-expression Network Analysis 18 (WGCNA) and multi-algorithm feature selection approaches to identify novel hub genes potentially involved in sepsis severity. The manuscript is well designed and the data are intriguing as novel genes have been identified. There are a few issues that need to be addressed to increase the impact of the study.
- The prediction nomogram used to identify the potential genes identifying sepsis severity is extended to suggest that these could be used as biomarkers. However, do any/all of these gene products circulate? This would be important information to include as a biomarker would be of little use if it cannot be readily measured. At the vey least the authors need to add a discussion of what is known regarding whether any of these genes are cell locked or do circulate. This would also increase the impact of the discussion.
- If any of these gene products circulate carrying out a study to validate the validity as a biomarker of sepsis severity would greatly increase the impact of the study.
- Failing 2. could the authors validate these genes I another sepsis transcriptomic data base? This would increase the confidence of the data presented which is entirely computational.
Minor issues
- The text in the figure panels are highly pixelated. The authors should try to improve the dpi of the figures.
Reviewer 2 Report
Comments and Suggestions for Authors
Dear author,
Sun et al. present the results of a transcriptomic analysis of sepsis patient sample timecourse. DEGs were identified, and affected pathways predicted by KEGG analysis. Gene Ontology analysis pointed to DEG involvement in complement activation and antigen binding. Predictive accuracy of the DEGs was high. DEGs were further correlated with clinical traits and disease progression over the sampling timecourse.
Comments
--------
1. More description of the patient cohort is required.
2. The integration of the GSE216902 (whole blood mRNA-seq) and GSE232753 (PBMCs RNA-seq) datasets raises concerns about potential batch effects due to differences in sample origins. While normalization/standardization was applied, biological variability between whole blood and PBMCs may confound results. Could the authors provide additional justification for combining these datasets and clarify steps taken to mitigate batch effects?
For Figure 1C, the Scale-Free Fit R² (0.6) falls below the recommended threshold of ≥0.8. Please clarify the rationale for selecting R²=0.6 as the cutoff and address whether the high mean connectivity (>100) reflects excessive outliers or unresolved batch effects.
3. The identification of TNFSF10, TMCC2, and PLVAP as key biomarkers is prefect. However, experimental validation (e.g., qPCR, Western blot, or functional assays in relevant sepsis models) would significantly enhance the translational relevance of these findings. Could you address plans or limitations for validating these candidates experimentally?
4. Figure Quality and Clarity. Several figures (1A, 1F, 3B, 5A/B/D) require improved resolution and labeling. Subtitles, p-values, and axis labels are indistinct, hindering interpretation.
5. Please specify the GSVA analysis statistical cutoffs (e.g., p-value and enrichment score thresholds) used to select pathways for figure 5A. Additionally, clarify whether the heatmap correlation values represents enrichment scores (ES).
6. The study’s workflow is logically structured, beginning with WGCNA integration of two sepsis datasets, followed by identification of 54 overlapping DEGs intersecting WGCNA and differential expression results. Subsequent refinement via machine learning (Random Forest, LASSO, SVM-RFE) narrows this to 10 high-confidence DEGs, culminating in ROC-based prioritization of TNFSF10, TMCC2, and PLVAP as candidate biomarkers.
However, Figure 5, which explores additional associations (gene correlations or pathway enrichments), currently disrupts the flow by appearing after the conclusion of the biomarker discovery storyline in Figure 4. Reordering Figure 5 after Figure 4 or integrating these results into the biomarker discovery storyline could be better.
7. I would like to see more explanation of the "clinical traits", or at least some examples, which we are told are correlated with DEGs. It is difficult to make any determination on their utility without more detail. The word "exacerbation" in the title is not strongly justified.
Regards
